# Learning to Learn Kernels with Variational Random Features

## Abstract

Meta-learning for few-shot learning involves a meta-learner that acquires shared knowledge from a set of prior tasks to improve the performance of a base-learner on new tasks with a small amount of data. Kernels are commonly used in machine learning due to their strong nonlinear learning capacity, which have not yet been fully investigated in the meta-learning scenario for few-shot learning. In this work, we explore kernel approximation with random Fourier features in the meta-learning framework for few-shot learning. We propose learning adapative kernels by meta variational random features (MetaVRF), which is formulated as a variational inference problem. To explore shared knowledge across diverse tasks, our MetaVRF deploys an LSTM inference network to generate informative features, which can establish kernels of highly representational power with low spectral sampling rates, while also being able to quickly adapt to specific tasks for improved performance. We evaluate MetaVRF on a variety of few-shot learning tasks for both regression and classification. Experimental results demonstrate that our MetaVRF can deliver much better or competitive performance than recent meta-learning algorithms.

## 1 Introduction

Humans have the instinct to effortlessly learn new concepts from a few examples and show great generalization ability to new samples. However, existing machine learning models, e.g., deep neural networks (DNNs) (Krizhevsky et al., 2012; He et al., 2016a), rely highly on large-scale annotated training data (Deng et al., 2009) to achieve satisfactory performance. The huge gap between human intelligence and DNNs motivates us to try and progress the task of learning from a few samples, a.k.a. few-shot learning (Fei-Fei et al., 2006; Lake et al., 2015; Ravi & Larochelle, 2017).

Learning to learn, or *meta-learning* (Schmidhuber, 1992), has recently received great interests in the machine learning community and offers a promising tool for few-shot learning (Andrychowicz et al., 2016; Ravi & Larochelle, 2017; Finn et al., 2017). Generally speaking, a meta-learner (Ravi & Larochelle, 2017; Bertinetto et al., 2019) is trained to improve the performance of a base-learner on individual tasks, which is also fast adapted to solve new tasks. The crux of meta-learning for few-shot learning is to explore the common knowledge, such as a good parameter initialization (Finn et al., 2017) or efficient optimization update rule (Andrychowicz et al., 2016; Ravi & Larochelle, 2017), shared across different tasks. The knowledge is accumulated and distilled throughout the learning stage, making the model adaptable to new but related tasks (Finn et al., 2017).

Kernel approximation by random Fourier features (RFFs) (Rahimi & Recht, 2007) is an effective technique for efficient kernel learning (Gärtner et al., 2002), which has recently become increasingly popular (Sinha & Duchi, 2016; Carratino et al., 2018). It resorts to the Fourier transform of shift-invariant kernels and constructs explicit feature maps using the Monte Carlo approximation of the Fourier representation. The desired kernel function is approximated by the inner products between these random features. Though demonstrating great potential as a strong base learner, kernel approximation with random features has not yet been fully explored in the meta-learning scenario for few-shot learning. It has already been shown that the classification performance of the kernel with random features does not correlate well with the accurate approximation of kernels. Learning adaptive kernels with random features, for instance, by data-driven sampling strategies (Sinha & Duchi, 2016), can improve the performance with a low sampling rate compared to using universal random features (Avron et al., 2016; Chang et al., 2017). However, since only a few samples are

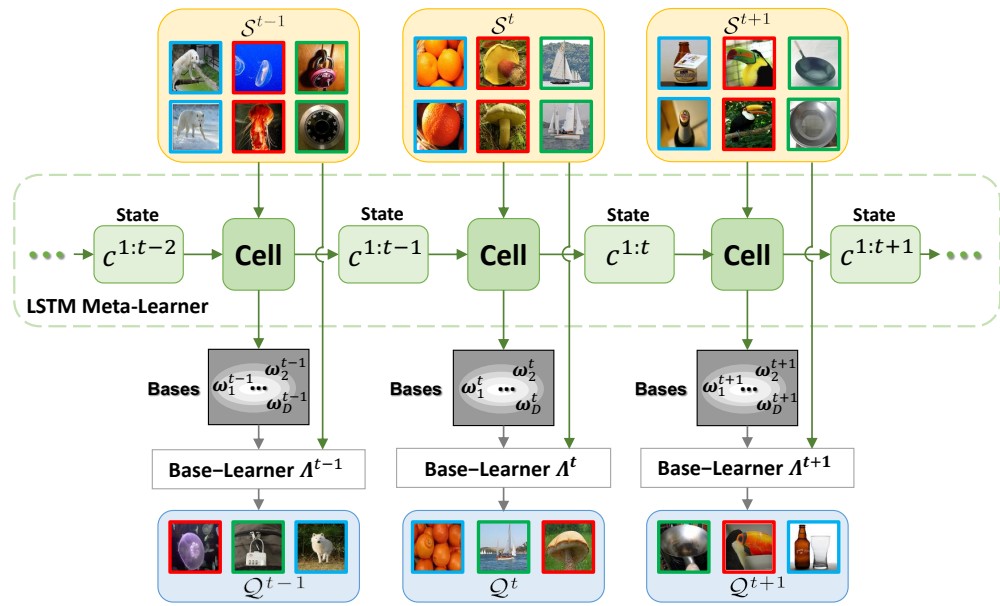

Figure 1: The inference framework of our meta variational random features (MetaVRF) with a long short-term memory (LSTM) network ($t$ is the task index). The meta-learner employs an LSTM network to infer random Fourier features from the support set $\mathcal{S}^t$ of the current task and bases $\boldsymbol{\omega}^{t-1}$ of previous tasks, which explores task dependency to extract shared knowledge. The base-learner is a classifier with kernels obtained by variational features from the meta-learner.

available in each task, it is challenging to learn adaptive kernels with data-driven random features while maintaining high representational capacity for few-shot learning tasks. To obtain powerful kernels for few-shot learning tasks, we need to fully explore the relationship among diverse tasks and capture their shared knowledge to generate informative random features.

In this work, we propose meta variational random features (MetaVRF) to approximate kernels in a data-driven manner for few-shot learning, which integrates variational inference and kernels in the meta-learning framework. Learning kernels with random Fourier features for few-shot learning allows us to leverage the universal approximation property of kernels to capture shared knowledge in related tasks, and meanwhile it enables us to learn adaptive basis functions to quickly and efficiently adapt to new tasks. Learning adaptive kernels with data-driven random features can be naturally cast into variational inference that approximates probability density through optimization, where the posterior over the random basis function is the spectral distribution of a translation-invariant kernel.

The inference of the posterior is conducted in the context of tasks to exploring their dependency for capturing shared knowledge. We adopt a long short-term memory (LSTM) based inference network (Hochreiter & Schmidhuber, 1997), which establishes task context inference to capture the task dependency. Specifically, during the inference, the cell state in the LSTM carries and accumulates the shared knowledge which is updated for each task throughout the course of learning. The *remember* and *forget* operations in the LSTM use new information to episodically refine the cell state by gaining experience from a batch of tasks, which can eventually produce random features of highly representational capability for all tasks. For an individual task, the task specific information is first extracted from the support set, and then combined with the shared knowledge in the shared cell state together as the joint condition, to infer the adaptive spectral distribution of the kernels. As a result, the task context inference can not only learn to extract and maintain the shared knowledge across tasks, but also leverage the task-specific knowledge to achieve an adaptive kernel to the current task. The inference framework of our MetaVRF is illustrated in Figure 1.

Extensive experiments on a variety of few-shot learning problems such as regression and classification demonstrate that, our MetaVRF method achieves competitive or even better performance when compared to state-of-the-art algorithms. Due to the advantages of kernels, our MetaVRF can be applied to test settings with different ways and shots from those of training setting, in which the promising results again validate the effectiveness of our MetaVRF for few-shot learning.

## 2 PROBLEM STATEMENT

In this section, we describe the setup of meta-learning for few-shot learning and introduce the kernel ridge regression as the base-learner, where kernels are approximated by random Fourier features.

### 2.1 META-LEARNING WITH KERNELS

We adopt the episodic training strategy (Ravi & Larochelle, 2017) commonly used for few-shot classification in meta-learning, which usually involves the *meta-training* and *meta-test* stages. In the *meta-training* stage, a meta-learner is trained to enhance the performance of a base-learner on a *meta-training* set with a batch of few-shot learning tasks, where a task is usually referred as an episode (Ravi & Larochelle, 2017). In the *meta-test* stage, the base learner is evaluated on a *meta-test* set with different classes of data samples from the *meta-training* set.

For the few-shot classification problem, we sample $C$-way $k$-shot classification tasks from the *meta-training* set, where $k$ is the number of labelled examples for each of the $C$ classes. Given the $t$-th task with a support set $\mathcal{S}^t = \{(\mathbf{x}_i, \mathbf{y}_i)\}_{i=1}^{C \times k}$ and query set $\mathcal{Q}^t = \{(\tilde{\mathbf{x}}_i, \tilde{\mathbf{y}}_i)\}_{i=1}^{m}$ ($\mathcal{S}^t, \mathcal{Q}^t \subseteq \mathcal{X}$), we learn the parameters $\alpha^t$ of the predictor $f_{\alpha^t}$ using a standard learning algorithm with kernel trick $\alpha^t = \Lambda(\Phi(X), Y)$, where $\mathcal{S}^t = \{X, Y\}$. Here, $\Lambda$ is the base-learner and $\Phi : \mathcal{X} \to \mathbb{R}^{\mathcal{X}}$ is a mapping function from $\mathcal{X}$ to a dot product space $\mathcal{H}$. The similarity measure $\mathbb{k}(\mathbf{x}, \mathbf{x}') = \langle \Phi(\mathbf{x}), \Phi(\mathbf{x}') \rangle$ is usually called a kernel (Hofmann et al., 2008).

In traditional supervised learning, the base-learner for the $t$-th single task usually uses a universal kernel to map the input onto a dot product space for efficient learning. Once the base-learner is trained on the support set, its performance is evaluated on the query set by the following loss function

$$\sum_{(\tilde{\mathbf{x}}, \tilde{\mathbf{y}}) \in \mathcal{Q}^t} L\left(f_{\alpha^t}\left(\Phi(\tilde{\mathbf{x}})\right), \tilde{\mathbf{y}}\right), \tag{1}$$

where $L(\cdot)$ can be any differentiable function, e.g., cross-entropy loss. In the meta-learning setting for few-shot learning, we usually consider a batch of tasks. Thus, the meta-learner is trained by optimizing the following objective function *w.r.t.* the empirical loss on $T$ tasks

$$\sum_{t} \sum_{(\tilde{\mathbf{x}}, \tilde{\mathbf{y}}) \in \mathcal{Q}^t} L\left(f_{\alpha^t}\left(\Phi^t(\tilde{\mathbf{x}})\right), \tilde{\mathbf{y}}\right), \quad \text{with} \quad \alpha^t = \Lambda\left(\Phi^t(X), Y\right), \tag{2}$$

where $\Phi^t$ is the feature mapping function which can be obtained by learning task-specific kernel $\mathbb{k}^t$ for each task $t$ with data-driven ramdom Fourier features.

In this work, we employ kernel ridge regression (KRR), which has an efficient closed-form solution, as the base-learner $\Lambda$ for few-shot learning. The kernel value in the Gram matrix $K \in \mathbb{R}^{Ck \times Ck}$ can be computed as $\mathbb{k}(\mathbf{x}, \mathbf{x}') = \Phi(\mathbf{x})\Phi(\mathbf{x}')^\top$, where "$\top$" is the transpose operation. The base-learner $\Lambda$ for a single task can be obtained by solving the following objective *w.r.t.* the support set of this task,

$$\Lambda = \arg\min_{\alpha} \text{Tr}[(Y - \alpha K)(Y - \alpha K)^\top] + \lambda \alpha K \alpha^\top. \tag{3}$$

This produces a closed-form solution $\alpha = (\lambda I + K)^{-1} Y$. The learned predictor is then applied to the query set for prediction of the query set $\tilde{X}$:

$$\hat{Y} = f_\alpha(\tilde{X}) = \alpha \tilde{K}, \tag{4}$$

where $\tilde{K} = \Phi(X)\Phi(\tilde{X})^\top \in \mathbb{R}^{Ck \times m}$ is with each element as $\mathbb{k}(\mathbf{x}, \tilde{\mathbf{x}})$ between the samples from the support and query sets. Note that we also treat $\lambda$ in Eq. (3) as a trainable parameter by leveraging the meta-learning setting, and all these parameters are learned by the meta-learner.

In order to obtain task-specific kernels, we propose to learn kernels with random Fourier features, which not only allows us to obtain task-adaptive kernels but also enables us to capture shared knowledge of different tasks by exploring their dependency.

### 2.2 RANDOM FOURIER FEATURES

Random Fourier features (RFFs) were proposed to construct explicit random feature maps using the Monte Carlo approximation of the Fourier representation (Rahimi & Recht, 2007), which is derived from Bochner's theorem (Rudin, 1962).

**Theorem 1 (Bochner's theorem)** *(Rudin, 1962) A continuous, real valued, symmetric and shift-invariant function* $\mathtt{k}(\mathbf{x}, \mathbf{x}') = \mathtt{k}(\mathbf{x} - \mathbf{x}')$ *on* $\mathbb{R}^d$ *is a positive definite kernel if and only if it is the Fourier transform* $p(\boldsymbol{\omega})$ *of a positive finite measure such that*

$$\mathtt{k}(\mathbf{x}, \mathbf{x}') = \int_{\mathbb{R}^d} e^{i\boldsymbol{\omega}^\top (\mathbf{x}-\mathbf{x}')} dp(\boldsymbol{\omega}) = \mathbb{E}_{\boldsymbol{\omega}}[\zeta_{\boldsymbol{\omega}}(\mathbf{x})\zeta_{\boldsymbol{\omega}}(\mathbf{x})^*], \quad where \quad \zeta_{\boldsymbol{\omega}}(\mathbf{x}) = e^{i\boldsymbol{\omega}^\top \mathbf{x}}. \tag{5}$$

It is guaranteed that $\zeta_{\boldsymbol{\omega}}(\mathbf{x})\zeta_{\boldsymbol{\omega}}(\mathbf{x})^*$ is an unbiased estimation of $\mathtt{k}(\mathbf{x}, \mathbf{x}')$ with sufficient RFF bases $\{\boldsymbol{\omega}\}$ drawn from $p(\boldsymbol{\omega})$ (Rahimi & Recht, 2007).

For a predefined kernel, e.g., radius basis function (RBF), we sample from its spectral distribution using the Monte Carlo method, and obtain the explicit feature map:

$$\mathbf{z}(\mathbf{x}) = \frac{1}{\sqrt{D}}[\cos(\boldsymbol{\omega}_1^\top \mathbf{x} + b_1), \cdots, \cos(\boldsymbol{\omega}_D^\top \mathbf{x} + b_D)], \tag{6}$$

where $\{\boldsymbol{\omega}_1, \cdots, \boldsymbol{\omega}_D\}$ are the random bases sampled from $p(\boldsymbol{\omega})$, and $[b_1, \cdots, b_D]$ are $D$ biases sampled from a uniform distribution with a range of $[0, 2\pi]$. Finally, the kernel value $\mathtt{k}(\mathbf{x}, \mathbf{x}') = \mathbf{z}(\mathbf{x})\mathbf{z}(\mathbf{x}')^\top$ in $K$ is computed as the dot product of their random feature maps with the same bases.

Learning adaptive kernel with data-driven random Fourier features is essential to find the posterior distribution and the specific spectral distribution of kernels. In the following section, we introduce our meta variational random features (MetaVRF), in which random Fourier bases are treated as latent variables inferred from the support set in the meta-learning setting.

## 3 META VARIATIONAL RANDOM FEATURES

### 3.1 META EVIDENCE LOWER BOUND

From the probabilistic perspective of view, the goal of few-shot learning is to maximize the conditional predictive log-likelihood of samples from the query set $\mathcal{Q}$. We treat the random Fourier base $\boldsymbol{\omega}$ of the kernel as a latent variable:

$$\max_p \sum_{(\mathbf{x},\mathbf{y})\in\mathcal{Q}} \log p(\mathbf{y}|\mathbf{x}, \mathcal{S}) = \max_p \sum_{(\mathbf{x},\mathbf{y})\in\mathcal{Q}} \log \int p(\mathbf{y}|\mathbf{x}, \mathcal{S}, \boldsymbol{\omega})p(\boldsymbol{\omega}|\mathbf{x}, \mathcal{S})d\boldsymbol{\omega}. \tag{7}$$

In order to infer the posterior $p(\boldsymbol{\omega}|\mathbf{y}, \mathbf{x}, \mathcal{S})$ over $\boldsymbol{\omega}$, which is generally intractable, we resort to using a variational distribution $q_\phi(\boldsymbol{\omega}|\mathcal{S})$ to approximate this posterior, where the base is conditioned on the support set $\mathcal{S}$ by leveraging meta-learning. We can obtain the variational distribution by minimizing the Kullback-Leibler (KL) divergence

$$D_{\mathrm{KL}}[q_\phi(\boldsymbol{\omega}|\mathcal{S})||p(\boldsymbol{\omega}|\mathbf{y}, \mathbf{x}, \mathcal{S})]. \tag{8}$$

By applying the Bayes' rule to the posterior $p(\boldsymbol{\omega}|\mathbf{y}, \mathbf{x}, \mathcal{S})$, we can derive the meta ELBO as

$$\log p(\mathbf{y}|\mathbf{x}, \mathcal{S}) \geq \sum_{(\mathbf{x},\mathbf{y})\in\mathcal{Q}} \mathbb{E}_{q_\phi(\boldsymbol{\omega}|\mathcal{S})} \log p(\mathbf{y}|\mathbf{x}, \mathcal{S}, \boldsymbol{\omega}) - D_{\mathrm{KL}}[q_\phi(\boldsymbol{\omega}|\mathcal{S})||p(\boldsymbol{\omega}|\mathbf{x}, \mathcal{S})] = \mathcal{L}_{\mathrm{MetaELBO}}.$$

$$\tag{9}$$

The first term of meta ELBO is the predictive log-likelihood conditioned on the observation $\mathbf{x}$, $\mathcal{S}$ and the inferred RFF bases $\boldsymbol{\omega}$. Maximizing it enables us to make an accurate prediction for the query set by utilizing the inferred bases from the support set. The second term in our meta ELBO minimizes the discrepancy between the meta variational distribution $q_\phi(\boldsymbol{\omega}|\mathcal{S})$ and the meta prior $p(\boldsymbol{\omega}|\mathbf{x}, \mathcal{S})$, which encourages samples from support and query sets to share the same random Fourier bases. The full derivation of the meta ELBO is provided in the Appendix B.

We now obtain the objective by maximizing the meta ELBO with respect to a batch of tasks:

$$\mathcal{L} = \frac{1}{T}\sum_{t=1}^{T} \left( \sum_{(\mathbf{x},\mathbf{y})\in\mathcal{Q}^t} \mathbb{E}_{q_\phi(\boldsymbol{\omega}^t|\mathcal{S}^t)} \log p(\mathbf{y}|\mathbf{x}, \mathcal{S}^t, \boldsymbol{\omega}^t) - D_{\mathrm{KL}}[q_\phi(\boldsymbol{\omega}^t|\mathcal{S}^t)||p(\boldsymbol{\omega}^t|\mathbf{x}, \mathcal{S}^t)] \right). \tag{10}$$

where $\mathcal{S}^t$ is the support set of the $t$-th task associated with its specific bases $\{\boldsymbol{\omega}_d^t\}_{d=1}^{D}$. Directly optimizing the above objective does not take into count the task dependency. We introduce task context inference by making the posterior conditioned on both the support set of the current task and the bases from previous tasks.

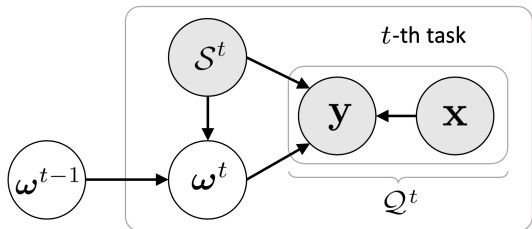

Figure 2: Illustration of MetaVRF in a directed graphical model. $(\mathbf{x}, \mathbf{y})$ is a test sample in the query set $\mathcal{Q}^t$. The base $\boldsymbol{\omega}^t$ is inferred by conditioning on both the base $\boldsymbol{\omega}^{t-1}$ from the previous task and the support set $\mathcal{S}^t$ of the current task.

### 3.2 TASK CONTEXT INFERENCE

To leverage the knowledge shared across tasks, we propose task context inference for random feature bases. Specifically, the bases $\{\boldsymbol{\omega}_d^t\}_{d=1}^D$ of the $t$-th task should rely on the base in the previous $t-1$ task, denoted as $\{\boldsymbol{\omega}_d^{t-1}\}_{d=1}^D$. The directed graphical model with related variables is shown in Figure 2. To compute the probability of $\boldsymbol{\omega}^t$ conditioned on $\boldsymbol{\omega}^{t-1}$, we replace the previous variational posterior $q_\phi(\boldsymbol{\omega}^t|\mathcal{S}^t)$ with $q_\phi(\boldsymbol{\omega}|\mathcal{S}^t, \boldsymbol{\omega}^{t-1})$. Therefore, the objective of the task context inference is:

$$\mathcal{L} = \frac{1}{T} \sum_{t=1}^T \left( \sum_{(\mathbf{x},\mathbf{y}) \in \mathcal{Q}^t} \mathbb{E}_{q_\phi(\boldsymbol{\omega}^t|\boldsymbol{\omega}^{t-1},\mathcal{S}^t)} \log p(\mathbf{y}|\mathbf{x}, \mathcal{S}^t, \boldsymbol{\omega}^t) - D_{\mathrm{KL}}[q_\phi(\boldsymbol{\omega}^t|\boldsymbol{\omega}^{t-1}, \mathcal{S}^t)||p(\boldsymbol{\omega}^t|\mathbf{x}, \mathcal{S}^t)] \right). \tag{11}$$

Note that the variational approximate posterior $q_\phi(\boldsymbol{\omega}^t|\boldsymbol{\omega}^{t-1}, \mathcal{S}^t)$ is a multivariate Gaussian with a diagonal covariance. Given the support set as input, the mean $\boldsymbol{\omega}_\mu$ and standard deviation $\boldsymbol{\omega}_\sigma$ are output from the LSTM inference network $\phi(\cdot)$. To enable the back-propagation of the LSTM inference network with the sampling operation during training, we leverage the reparametrization trick (Kingma & Welling, 2013) as

$$\boldsymbol{\omega}^{(l)} = \boldsymbol{\omega}_\mu + \boldsymbol{\omega}_\sigma \odot \boldsymbol{\epsilon}^{(l)} \quad \text{with} \quad \boldsymbol{\epsilon}^{(l)} \sim \mathcal{N}(0, \mathrm{I}). \tag{12}$$

We use a permutation-invariant instance pooling layer to aggregate the support set $\mathcal{S}^t$ of a group of examples, which essentially takes the average over the feature vectors of samples in the support set into a single vector, as in (Zaheer et al., 2017). In practice, the feature representation $\mathbf{e}$ is extracted for each image $\mathbf{x}$ by a shared convolutional network $\psi(\cdot)$, i.e., $\mathbf{e} = \psi(\mathbf{x})$. The aggregation of samples in the support set is denoted as $\bar{\mathbf{e}}$.

We propose an LSTM-type inference network inspired by the fact that the long-term memory can be carried and refined in cell states $\mathbf{c}$ during its update (Gers & Schmidhuber, 2000). Specifically, we design a simplified variant of LSTM with two gates to remove the effect of the short-term memory. The common knowledge shared by tasks is stored in the cell state and updated with new information in each episode. During inference, LSTM can remove trivial information and add knowledge with a highly representational ability to the cell state. Once we have updated the cell state, the shared knowledge stored within it is combined with the task-specific information from the input support set to infer the spectral distribution. The update steps in the LSTM network are

$$\begin{aligned}
\boldsymbol{f}^t &= \mathrm{sigmoid}(W_f \cdot [\bar{\mathbf{e}}^t, \mathbf{c}^{t-1}] + \mathbf{b}_f); \\
\boldsymbol{i}^t &= \mathrm{sigmoid}(W_i \cdot [\bar{\mathbf{e}}^t, \mathbf{c}^{t-1}] + \mathbf{b}_i); \\
\hat{\mathbf{c}}^t &= \tanh(W_c \cdot [\bar{\mathbf{e}}^t] + \mathbf{b}_c); \\
\mathbf{c}^t &= \boldsymbol{f}^t \cdot \mathbf{c}^{t-1} + \boldsymbol{i}^t \cdot \hat{\mathbf{c}}^t; \\
\boldsymbol{\omega}_\mu^t &= \tanh(W_o \cdot [\bar{\mathbf{e}}^t, \mathbf{c}^t] + \mathbf{b}_o).
\end{aligned} \tag{13}$$

$\boldsymbol{\omega}_\sigma$ is computed in the same way. After training, the final state $\mathbf{c}^T$ is the shared knowledge from the *meta-training* set, which is directly used in the *meta-testing* set. In the *meta-testing* stage, we use the LSTM inference network $\phi$ to obtain $\boldsymbol{\omega}_\mu$ and $\boldsymbol{\omega}_\sigma$ from $\mathcal{S}$ and $\boldsymbol{\omega}^T$. Then, a set of bases are sampled from $q_\phi(\boldsymbol{\omega}|\boldsymbol{\omega}^T, \mathcal{S})$ to construct the random features in the kernels. Finally, the base-learner is optimized to obtain the predictor for testing.

## 4 RELATED WORK

**Meta-learning**, or learning to learn, endues machine learning models the ability to improve their performance with for a number of training tasks. It has received increasing research interest with breakthroughs in many directions (Finn et al., 2017; Rusu et al., 2019; Gordon et al., 2019; Aravind Rajeswaran, 2019). Metric-based methods cast few-shot learning as a matching problem (Vinyals et al., 2016; Snell et al., 2017; Sung et al., 2018; Satorras & Estrach, 2018; Allen et al., 2019) by learning a shared similarity function for diverse tasks. Graphical neural network (GNN) based model generalizes the matching methods by learning the message propagation from the support set and transfer it to the query set (Garcia & Bruna, 2018). Gradient-based methods (e.g., MAML (Finn et al., 2017)) learn an appropriate initialization of model parameters and adapt it on new tasks with only a few gradient steps (Finn & Levine, 2018; Zintgraf et al., 2019; Rusu et al., 2019). Some other interesting works propose to directly learn the gradient optimization process of networks using RNNs (Ravi & Larochelle, 2017; Andrychowicz et al., 2016). Memory-based methods learn to leverage an external memory module to store and leverage key knowledge for quick adaptation (Santoro et al., 2016; Ramalho & Garnelo, 2019). Bayesian meta-learning methods (Edwards & Storkey, 2017; Finn et al., 2018; Gordon et al., 2019) usually rely on hierarchical Bayesian models to learn the shared statistical information among different tasks and reason about the uncertainty over models. Differentiable solution methods (Liu et al., 2019; Bertinetto et al., 2019) learn a universal feature embedding and obtain a task-specific learner with a closed-form solution.

While those meta-learning algorithms have achieved great success in solving few-shot learning tasks, it remains an open challenge to explore shared knowledge across prior tasks. In this work, we introduce kernel learning with data-driven random Fourier features to explore task dependency to extract the shared knowledge.

**Kernel learning with random Fourier features** is a versatile and powerful tool in the machine learning and statistics communities (Bishop, 2006; Hofmann et al., 2008; Shervashidze et al., 2011). Pioneering works in this category (Bach et al., 2004; Gönen & Alpaydın, 2011; Duvenaud et al., 2013) learn to combine predefined kernels in a multi-kernel learning manner. As an innovative feature map, several studies have focused on random Fourier features (RFFs) (Rahimi & Recht, 2007), with recent works (Wilson & Adams, 2013) learning kernels in the frequency domain by modeling the spectral distribution as a mixture of Gaussians and computing its optimal linear combination. Instead of modeling the spectral distribution with explicit density functions, other works focus on optimizing the random base sampling strategy (Yang et al., 2015; Sinha & Duchi, 2016).

To the best of our knowledge, our work is the first to extend kernel learning with random features to the meta-learning framework for few-shot learning. We propose training a meta-learner to infer the spectral distribution of random features from the support set. The task-specific kernel can be leveraged by the base-learner for the supervised learning. Compared with RFFs, our learned MetaVRF achieves superior performance on few-shot learning tasks with a low sampling rate of bases.

## 5 EXPERIMENTS

In this section, we evaluate our MetaVRF on several few-shot learning problems for both regression and classification. We conduct classification experiments on three commonly-used benchmark datasets, i.e., Omniglot (Lake et al., 2015), miniImageNet (Vinyals et al., 2016) and CIFAR-FS (Krizhevsky et al., 2009). More details about the three datasets are provided in the Appendix C. We also conduct deeper analysis to validate the effectiveness of our MetaVRF.

### 5.1 FEW-SHOT REGRESSION

We begin with a $k$-shot regression problem, and compare our MetaVRF with MAML (Finn et al., 2017) as the baseline method. We follow the MAML work (Finn et al., 2017) to fit a target sine function $y = A\sin(wx + b)$, with only a few annotated samples. $A \in [0.1, 5]$, $w \in [0.8, 1.2]$, and $b \in [0, \pi]$ denote the amplitude, frequency, and phase, respectively, which follow a uniform distribution within the corresponding interval. The goal is to estimate the target sine function given only $n$ randomly sampled date points. In our experiments, we consider input in the range of $x \in [-5, 5]$, and conduct three tests under the conditions of $k = 3, 5, 10$. For a fair comparison, we compute the feature

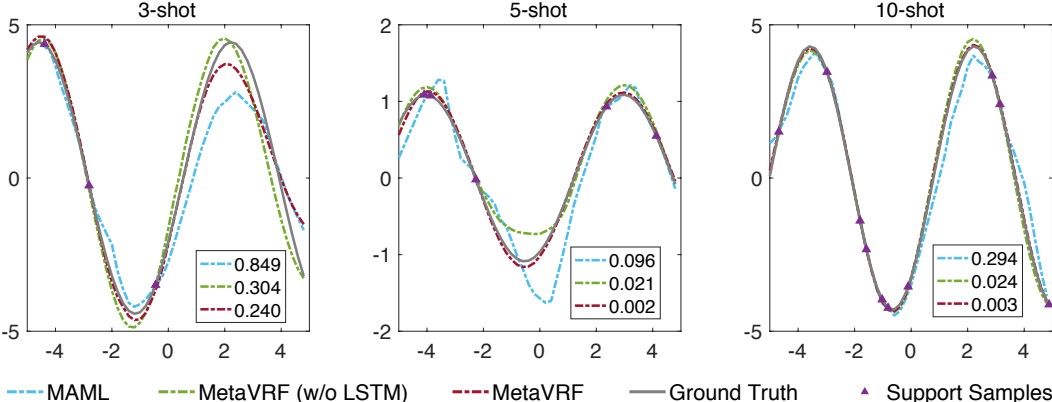

Figure 3: Performance comparison for few-shot regression. Values are in term of MSE between three methods and ground truth.

embedding using a small multi-layer perceptron (MLP) with two hidden layers of size 40, following the same settings as used in MAML. We have also experimented with our MetaVRF without using LSTM (MetaVRF w/o LSTM) for the inference.

The comparison results are plotted in Figure 3. The proposed MetaVRF can fit the function accurately with even three shots. With the increase of shots, our MetaVRF can perform better, almost entirely fitting the target function with 10 shots. It clearly shows that the performance can be improved by adopting the LSTM inference, which verifies the benefits of exploring task dependency. In addition, we can see that our MetaVRF performs much better than MAML which is a representative meta-learning algorithm for all three settings with different numbers of shots. More few-shot regression results are provided in the Appendix E.

## 5.2 FEW-SHOT CLASSIFICATION

For the classification task, we compare the proposed MetaVRF with the baseline method, random Fourier features (RFFs) and other state-of-the-art models. In particular, RFFs are the random feature computed from the universal kernel without adaptation.

**Experimental settings.** Image features are extracted via a shallow convolutional neural network. We employ the same architecture used in (Gordon et al., 2019). We do not use any fully connected layer for these CNNs. The dimension of all feature vectors is 256. The inference network for RFFs is replaced with a feed-forward network with a couple of dense blocks. The key hyperparameter of the number of bases $D$ in Eq. (6), is set to $D = 512$ for MetaVRF in all experiments, while we set $D = 2048$ for RFFs. The sampling rate of MetaVRF is much lower than in previous studies on RFFs, in which $D$ is usually set to be 5 to 10 times the dimension of the input features (Yu et al., 2016; Rahimi & Recht, 2007). All details are provided in the Appendix D.

**Quantitive analysis.** We evaluate our MetaVRF on three few-shot classification benchmarks using the same meta-testing protocol as (Gordon et al., 2019), and present the results with 95% confidence intervals. All reported results are for models trained from scratch for few-shot learning. Table 1 and 2 report the performance of MetaVRF compared to the current state of the arts with shallow CNN architectures. Our MetaVRF method achieves the new state-of-the-art results on most of the challenging datasets, (*e.g.*, 54.3% for 5-way 1-shot on *mini*ImageNet, which is a 1% improvement over the second-best method). On the Omniglot dataset, previous approaches did not specify the splits for training, validation, and testing, which may result in unfair comparisons. Our MetaVRF method falls within the error bars of the state-of-the-art models on all experiments under 5-way 1-shot, 5-way 5-shot, 20-way 1-shot and 20-way 5-shot settings.

Since the feature extraction network used in our MetaVRF is different from that used in previous works, in Table 2, we show the results of methods that have the same training procedures and conventional CNN architectures as ours. In Vinyals et al. (2016); Snell et al. (2017); Ravi & Larochelle (2017); Finn et al. (2017); Sung et al. (2018), they usually use 64 filters in each convolutional layer. Ravi & Larochelle (2017); Finn et al. (2017) set the number of filters to 32 for *mini*ImageNetto avoid

Table 1: Classification accuracies (%) on *mini*ImageNet and CIFAR-FS.

| Method | *mini*ImageNet, 5-way | | CIFAR-FS, 5-way | |
|---|---|---|---|---|
| | 1-shot | 5-shot | 1-shot | 5-shot |
| MATCHING NET (Vinyals et al., 2016) | 44.2 | 57 | — | — |
| MAML (Finn et al., 2017) | 48.7±1.8 | 63.1±0.9 | 58.9±1.9 | 71.5±1.0 |
| MAML (64C) | 46.7±1.7 | 61.1±0.1 | 58.9±1.8 | 71.5±1.1 |
| META-LSTM (Ravi & Larochelle, 2017) | 43.4±0.8 | 60.6±0.7 | — | — |
| PROTO NET (Snell et al., 2017) | 47.4±0.6 | 65.4±0.5 | 55.5±0.7 | 72.0±0.6 |
| RELATION NET (Sung et al., 2018) | 50.4±0.8 | 65.3±0.7 | 55.0±1.0 | 69.3±0.8 |
| SNAIL (32C) by (Bertinetto et al., 2019) | 45.1 | 55.2 | — | — |
| GNN (Garcia & Bruna, 2018) | 50.3 | 66.4 | 61.9 | 75.3 |
| PLATIPUS (Finn et al., 2018) | 50.1±1.9 | — | — | — |
| VERSA (Gordon et al., 2019) | 53.3±1.8 | 67.3±0.9 | 62.5±1.7 | 75.1±0.9 |
| R2-D2* (Bertinetto et al., 2019) | 50.5±0.2 | 65.4±0.2 | 62.3±0.2 | 77.4±0.2 |
| R2-D2 (Devos et al., 2019) | 51.7±1.8 | 63.3±0.9 | 60.2±1.8 | 70.9±0.9 |
| CAVIA (Zintgraf et al., 2019) | 51.8±0.7 | 65.6±0.6 | — | — |
| IMAML (Aravind Rajeswaran, 2019) | 49.3±1.9 | — | — | — |
| RFFS (2048d) | 54.0±1.9 | 65.4±0.9 | 61.3±1.8 | 75.1±0.9 |
| METAVRF (w/o LSTM, 512d) | 52.9±1.8 | 67.3±0.9 | 62.3±1.8 | 75.9 ±0.9 |
| METAVRF (512d) | **54.3**±1.9 | **68.0**±0.9 | **62.9**±0.7 | **76.3**±0.3 |

*training with 20 ways, test on 5 ways.

Table 2: Classification accuracies (%) on Omniglot.

| Method | Omniglot, 5-way | | Omniglot, 20-way | |
|---|---|---|---|---|
| | 1-shot | 5-shot | 1-shot | 5-shot |
| SIAMESE NET (Koch, 2015) | 96.7 | 98.4 | 88 | 96.5 |
| MATCHING NET (Vinyals et al., 2016) | 98.1 | 98.9 | 93.8 | 98.5 |
| MAML (Finn et al., 2017) | 98.7±0.4 | **99.9**±0.1 | 95.8±0.3 | 98.9±0.2 |
| PROTO NET (Snell et al., 2017) | 98.5±0.2 | 99.5±0.1 | 95.3±0.2 | 98.7±0.1 |
| SNAIL (Mishra et al., 2018) | 99.1±0.2 | 99.8 ±0.1 | **97.6** ±0.3 | **99.4** ±0.2 |
| GNN (Garcia & Bruna, 2018) | 99.2 | 99.7 | 97.4 | **99.0** |
| VERSA (Gordon et al., 2019) | **99.7**±0.2 | 99.8±0.1 | **97.7**±0.3 | 98.8±0.2 |
| R2-D2 (Bertinetto et al., 2019) | 98.6 | 99.7 | 94.7 | 98.9 |
| IMP (Allen et al., 2019) | 98.4±0.3 | 99.5±0.1 | 95.0±0.1 | 98.6±0.1 |
| RFFS (2048d) | 99.5±0.2 | 99.5±0.2 | 97.2±0.3 | 98.3±0.2 |
| METAVRF (w/o LSTM, 512d) | 99.6±0.2 | 99.6±0.2 | 97.0±0.3 | 98.4±0.2 |
| METAVRF (512d) | **99.8**±0.2 | **99.8**±0.1 | **97.5**±0.3 | **99.0**±0.2 |

overfitting. For a fair comparison, we increase the number of filters in MAML from 32 to 64 but obtains the inferior performance, indicating that MAML might be heavily prone to overfitting as the model size increases. In original SNAIL, they use the deep ResNet-12 embedding. We present the result of SNAIL reported in Bertinetto et al. (2019), where the result is obtained using similar shallow networks as ours. The graph neural network (GNN) (Garcia & Bruna, 2018) leverages a large embedding network with [64, 96, 128, 256] filters and one fully connected layer. For R2-D2 (Bertinetto et al., 2019), the numbers of filters are [96, 192, 384, 512] and we present the results of two of its variants with 64 channels. Different training and testing conditions may also affect the performance of the compared methods. We maintain consistent conditions to be consistent in our experiments but present the results of R2-D2 (Bertinetto et al., 2019) trained in 20 ways for 5-way tasks. The performance of the reproduced R2-D2 (Devos et al., 2019) is heavily impaired when this strategy is not employed.

In addition, recent interesting works of (Rusu et al., 2019; Gidaris & Komodakis, 2019; Li et al., 2019; Qiao et al., 2018; Gidaris & Komodakis, 2018) are not included for comparison because they rely on pre-trained embeddings or large-scale deep architectures, e.g., ResNet (He et al., 2016b). In contrast, we adopt a relatively shallow convolutional architecture for feature extraction to demonstrate the effectiveness of the proposed MetaVRF instead of relying on huge, powerful convolutional networks.

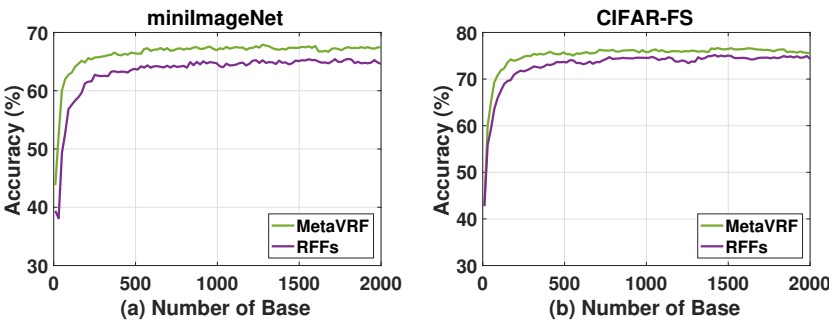

Figure 4: Performance with different numbers $D$ of bases.

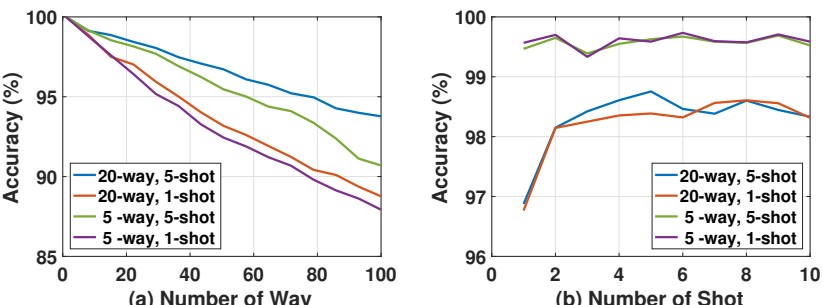

Figure 5: Performance on test tasks with varied ways and shots on Omniglot.

### 5.3 FURTHER ANALYSIS

**Efficiency.** In Figure 4, we demonstrate the effectiveness on bases of our MetaVRF with a low sampling rate by comparing it with conventional random Fourier features (RFFs) . We present the mean testing accuracy of fully trained models *w.r.t.* RFFs and our MetaVRF as the number $D$ of bases in Eq. (6) varies. Given the same number of sampled bases, our MetaVRF method consistently achieves much higher performance than RFFs, e.g., as shown for the 5-way, 5-shot condition in Figure 4. The results verify the effectiveness of our MetaVRF in learning adaptive kernels and exploring the tasks dependencies.

**Versatility.** Our MetaVRF also shows promising performance when the number of ways $C$ and $k$ shots between training and testing are inconsistent. In Figure 5, we plot the testing accuracy of the trained models on one particular $C$-way-$k$-shot task, with varied $C$ and $k$ in the testing stage. The results demonstrate that the trained model can maintain reasonable discriminability for a high number of testing ways. In particular, the model trained for the 20-way-5-shot task can retain high accuracy of $94\%$ when tested under the 100-way condition, as shown in Figure 5(a). The results also indicate that our MetaVRF exhibits considerable robustness and flexibility to a great variety of testing conditions.

## 6 CONCLUSIONS

In this paper, we explore kernel approximation based on random Fourier features in the meta-learning framework for few-shot learning. We propose the novel meta variational random features (MetaVRF), which leverages variational inference and meta-learning to infer the spectral distribution of random Fourier features in a data-driven way. MetaVRF can generate random Fourier features of high representational power and a relatively low spectral sampling rate by using an LSTM inference network to explore the shared knowledge. In practice, our LSTM inference network demonstrates the great ability to quickly adapt to specific tasks for improved performance. Experiments on few-shot learning tasks for several benchmark datasets demonstrate the state-of-the-art performance over previous methods and the importance of our contribution.

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

## A    RESULTS OF DEEP ARCHITECTURES

Table 3: Classification accuracies (%) on *mini*ImageNet with 5 way using deep architectures

| Method | 1-shot | 5-shot |
|---|---|---|
| **META-SGD** (Li et al., 2017) | 54.24±0.03 | 70.86±0.04 |
| (**GIDARIS & KOMODAKIS, 2018**) | 56.20±0.86 | 73.00±0.64 |
| (**BAUER ET AL., 2017**) | 56.30±0.40 | 73.90±0.30 |
| (**MUNKHDALAI ET AL., 2017**) | 57.10±0.70 | 70.04±0.63 |
| (**QIAO ET AL., 2018**) | 59.60±0.41 | 73.54±0.19 |
| **LEO** (Rusu et al., 2019) | 61.76±0.08 | 77.59±0.12 |
| **SNAIL** (Mishra et al., 2018) | 55.71±0.99 | 68.88±0.92 |
| **TADAM** (Oreshkin et al., 2018) | 58.50±0.30 | 76.70±0.30 |
| **METAVRF** (ours) | **63.40**±0.06 | **77.85**±0.28 |

We evaluate our method by using pre-trained embeddings and compare with latest methods with deep embedding architectures. We use the pre-trained embeddings from a 28-layer Wide Residual Network (WRN-28-10) (Zagoruyko & Komodakis, 2016), in a similar fashion to (Rusu et al., 2019; Bauer et al., 2017; Qiao et al., 2018). Specifically, the WRN-28-10 was trained to classify images which are all from the meta-training set into the corresponding categories. We select activations in the 21th layer, with average pooling over spatial dimensions, as feature embeddings. The dimension of pre-trained embeddings is 640. As shown in Table. 3, our MetaVRF achieves the state-of-the-art preformance on both cases of 1-shot and 5-shot.

## B    DERIVATIONS OF META ELBO

For a singe task, we begin with maximizing log-likelihood of the conditional distribution $p(\mathbf{y}|\mathbf{x}, \mathcal{S})$ to derive the ELBO of MetaVRF. By leveraging Jensen's inequality, we have the following steps as

$$\log p(\mathbf{y}|\mathbf{x}, \mathcal{S}) = \log \int p(\mathbf{y}|\mathbf{x}, \mathcal{S}, \boldsymbol{\omega}) p(\boldsymbol{\omega}|\mathbf{x}, \mathcal{S}) d\boldsymbol{\omega} \tag{14}$$

$$= \log \int p(\mathbf{y}|\mathbf{x}, \mathcal{S}, \boldsymbol{\omega}) p(\boldsymbol{\omega}|\mathbf{x}, \mathcal{S}) \frac{q_\phi(\boldsymbol{\omega}|\mathcal{S})}{q_\phi(\boldsymbol{\omega}|\mathcal{S})} d\boldsymbol{\omega} \tag{15}$$

$$\geq \int \log \left[ \frac{p(\mathbf{y}|\mathbf{x}, \mathcal{S}, \boldsymbol{\omega}) p(\boldsymbol{\omega}|\mathbf{x}, \mathcal{S})}{q_\phi(\boldsymbol{\omega}|\mathcal{S})} \right] q_\phi(\boldsymbol{\omega}|\mathcal{S}) d\boldsymbol{\omega} \tag{16}$$

$$= \underbrace{\mathbb{E}_{q_\phi(\boldsymbol{\omega}|\mathcal{S})} \log \left[ p(\mathbf{y}|\mathbf{x}, \mathcal{S}, \boldsymbol{\omega}^t) \right] - D_{\mathrm{KL}}[q_\phi(\boldsymbol{\omega}|\mathcal{S})||p(\boldsymbol{\omega}|\mathbf{x}, \mathcal{S})]}_{\text{Meta ELBO}}. \tag{17}$$

We can also derive Meta ELBO from the KL divergence between the posterior $p(\boldsymbol{\omega}|\mathbf{y}, \mathbf{x}, \mathcal{S})$ and its variational posterior $q_\phi(\boldsymbol{\omega}|\mathcal{S})$ as following

$$D_{\mathrm{KL}}[q_\phi(\boldsymbol{\omega}|\mathcal{S})||p(\boldsymbol{\omega}|\mathbf{y}, \mathbf{x}, \mathcal{S})] \tag{18}$$

$$= \mathbb{E}_{q_\phi(\boldsymbol{\omega}|\mathcal{S})} \left[ \log q_\phi(\boldsymbol{\omega}|\mathcal{S}) - \log p(\boldsymbol{\omega}|\mathbf{y}, \mathbf{x}, \mathcal{S}) \right] \tag{19}$$

$$= \mathbb{E}_{q_\phi(\boldsymbol{\omega}|\mathcal{S})} \left[ \log q_\phi(\boldsymbol{\omega}|\mathcal{S}) - \log \frac{p(\mathbf{y}|\boldsymbol{\omega}, \mathbf{x}, \mathcal{S}) p(\boldsymbol{\omega}|\mathbf{x}, \mathcal{S})}{p(\mathbf{y}|\mathbf{x}, \mathcal{S})} \right] \tag{20}$$

$$= \log p(\mathbf{y}|\mathbf{x}, \mathcal{S}) + \mathbb{E}_{q_\phi(\boldsymbol{\omega}|\mathcal{S})} \left[ \log q_\phi(\boldsymbol{\omega}|\mathcal{S}) - \log p(\mathbf{y}|\boldsymbol{\omega}, \mathbf{x}, \mathcal{S}) - \log p(\boldsymbol{\omega}|\mathbf{x}, \mathcal{S}) \right] \tag{21}$$

$$= \log p(\mathbf{y}|\mathbf{x}, \mathcal{S}) - \mathbb{E}_{q_\phi(\boldsymbol{\omega}|\mathcal{S})} \left[ \log p(\mathbf{y}|\boldsymbol{\omega}, \mathbf{x}, \mathcal{S}) \right] + D_{\mathrm{KL}}[q_\phi(\boldsymbol{\omega}|\mathcal{S})||p(\boldsymbol{\omega}|\mathbf{x}, \mathcal{S})] \tag{22}$$

$$\geq 0. \tag{23}$$

Therefore, the lower bound of the evidence $p(\mathbf{y}|\mathbf{x})$ is at the RHS of

$$\log p(\mathbf{y}|\mathbf{x}, \mathcal{S}) \geq \mathbb{E}_{q_\phi(\boldsymbol{\omega}|\mathcal{S})} \log \left[ p(\mathbf{y}|\mathbf{x}, \mathcal{S}, \boldsymbol{\omega}^t) \right] - D_{\mathrm{KL}}[q_\phi(\boldsymbol{\omega}|\mathcal{S})||p(\boldsymbol{\omega}|\mathbf{x}, \mathcal{S})], \tag{24}$$

which is consistent with Eq. (17).

Table 4: The fully connected network $\psi(\cdot)$ used for regression.

| Output size | Layers |
|---|---|
| 1 | Input training samples |
| 40 | fully connected, RELU |
| 40 | fully connected, RELU |

Table 5: The CNN architecture $\psi(\cdot)$ for Omniglot.

| Output size | Layers |
|---|---|
| $28 \times 28 \times 1$ | Input images |
| $14 \times 14 \times 64$ | *conv2d* ($3 \times 3$, stride=1, SAME, RELU), dropout 0.9, *pool* ($2 \times 2$, stride=2, SAME) |
| $7 \times 7 \times 64$ | *conv2d* ($3 \times 3$, stride=1, SAME, RELU), dropout 0.9, *pool* ($2 \times 2$, stride=2, SAME) |
| $4 \times 4 \times 64$ | *conv2d* ($3 \times 3$, stride=1, SAME, RELU), dropout 0.9, *pool* ($2 \times 2$, stride=2, SAME) |
| $2 \times 2 \times 64$ | *conv2d* ($3 \times 3$, stride=1, SAME, RELU), dropout 0.9, *pool* ($2 \times 2$, stride=2, SAME) |
| 256 | flatten |

## C  FEW-SHOT CLASSIFICATION DATASETS

**Omniglot** (Lake et al., 2015) is a benchmark of few-shot learning that contain 1623 handwritten characters (each with 20 examples). All characters are grouped in 50 alphabets. For fair comparison against the state of the arts, we follow the same data split and pre-processing used in Vinyals et al. (2016). The training, validation, and testing are composed of a random split of $[1100, 200, 423]$. The dataset is augmented with rotations of 90 degrees, which results in 4000 classes for training, 400 for validation, and 1292 for testing. The number of examples is fixed as 20. All images are resized to $28 \times 28$. For a $C$-way, $k$-shot task at training time, we randomly sample $C$ classes from the 4000 classes. Once we have $C$ classes, $(k + 15)$ examples of each are sampled. Thus, there are $C \times k$ examples in the support set and $C \times 15$ examples in the query set. The same sampling strategy is also used in validation and testing.

***mini*ImageNet** (Vinyals et al., 2016) is a challenging dataset constructed from ImageNet (Russakovsky et al., 2015), which comprises a total of 100 different classes (each with 600 instances). All these images have been downsampled to $84 \times 84$. We use the same splits of Ravi & Larochelle (2017), where there are $[64, 16, 20]$ classes for training, validation and testing. We use the same episodic manner as Omniglot for sampling.

**CIFAR-FS** (CIFAR100 few-shots) (Bertinetto et al., 2019) is adapted from the CIFAR-100 dataset (Krizhevsky et al., 2009) for few-shot learning. Recall that in the image classification benchmark CIFAR-100, there are 100 classes grouped into 20 superclasses (each with 600 instances). CIFAR-FS use the same split criteria $(64, 16, 20)$ with which *mini*ImageNet has been generated. The resolution of all images is $32 \times 32$.

## D  MORE EXPERIMENTAL DETAILS

We train all models using the Adam optimizer (Kingma & Ba, 2014) with a learning rate of 0.0001. The other training setting and network architecture for regression and classification on three datasets are different as following.

### D.1  FEATURE EMBEDDING NETWORKS

**Regression.** The fully connected architecture for regression tasks is shown in Table 4. We train all three models (3-shot, 5-shot, 10-shot) over a total of $20,000$ iterations, with 6 episodes per iteration.

**Classification.** The CNN architectures for Omniglot, CIFAR-FS, and *mini*ImageNet are shown in Table 5, 6, and 7.

Table 6: The CNN architecture $\psi(\cdot)$ for CIFAR-FS

| Output size | Layers |
|---|---|
| 32×32×3 | Input images |
| 16×16×64 | *conv2d* (3×3, stride=1, SAME, RELU), dropout 0.5, *pool* (2×2, stride=2, SAME) |
| 8×8×64 | *conv2d* (3×3, stride=1, SAME, RELU), dropout 0.5, *pool* (2×2, stride=2, SAME) |
| 4×4×64 | *conv2d* (3×3, stride=1, SAME, RELU), dropout 0.5, *pool* (2×2, stride=2, SAME) |
| 2×2×64 | *conv2d* (3×3, stride=1, SAME, RELU), dropout 0.5, *pool* (2×2, stride=2, SAME) |
| 256 | flatten |

Table 7: The CNN architecture $\psi(\cdot)$ for *mini*ImageNet

| Output size | Layers |
|---|---|
| 84×84×3 | Input images |
| 42×42×64 | *conv2d* (3×3, stride=1, SAME, RELU), dropout 0.5, *pool* (2×2, stride=2, SAME) |
| 21×21×64 | *conv2d* (3×3, stride=1, SAME, RELU), dropout 0.5, *pool* (2×2, stride=2, SAME) |
| 10×10×64 | *conv2d* (3×3, stride=1, SAME, RELU), dropout 0.5, *pool* (2×2, stride=2, SAME) |
| 5×5×64 | *conv2d* (3×3, stride=1, SAME, RELU), dropout 0.5, *pool* (2×2, stride=2, SAME) |
| 2×2×64 | *conv2d* (3×3, stride=1, SAME, RELU), dropout 0.5, *pool* (2×2, stride=2, SAME) |
| 256 | flatten |

### D.2 INFERENCE NETWORKS

The architecture of the inference network for the regression task is in Table 8. For few-shot classification tasks, all models share the same architecture, as in Table 9.

### D.3 PRIOR NETWORKS

The architecture of the prior network for the regression task is in Table 10. For few-shot classification tasks, all models share the same architecture, as in Table 11.

### D.4 OPTIMIZATION SETTINGS

The number of training iterations and the batch size (episodes per iteration) are listed in Table 12.

Table 8: The inference network $\phi(\cdot)$ used for regression.

| Output size | Layers |
|---|---|
| 40 | Input samples feature |
| 40 | fully connected, ELU |
| 40 | fully connected, ELU |
| 40 | LSTM cell, Tanh to $\mu_w$, $\log \sigma_w^2$ |

Table 9: The inference network $\phi(\cdot)$ used for Omniglot, *mini*ImageNet, CIFAR-FS

| Output size | Layers |
|---|---|
| $k \times 256$ | Input feature |
| 256 | instance pooling |
| 256 | fully connected, ELU |
| 256 | fully connected, ELU |
| 256 | fully connected, ELU |
| 256 | LSTM cell, tanh to $\mu_w$, $\log \sigma_w^2$ |

Table 10: The prior network used for regression.

| Output size | Layers |
|---|---|
| 80 | The concatenation of query feature and aggregated support features |
| 40 | fully connected, ELU |
| 40 | fully connected, ELU |
| 40 | fully connected to $\mu_w$, $\log \sigma_w^2$ |

Table 11: The prior network used for Omniglot, *mini*ImageNet, CIFAR-FS

| Output size | Layers |
|---|---|
| 512 | The concatenation of query feature and aggregated support features |
| 256 | instance pooling |
| 256 | fully connected, ELU |
| 256 | fully connected, ELU |
| 256 | fully connected to $\mu_w$, $\log \sigma_w^2$ |

Table 12: Iteration and batch size for all datasets.

| Dataset | Iter. | Batch size |
|---|---|---|
| Regression | $20,000$ | 25 |
| Omniglot | $100,000$ | 6 |
| CIFAR-FS | $200,000$ | 8 |
| *mini*ImageNet | $100,000$ | 6 |

# E  FEW-SHOT REGRESSION

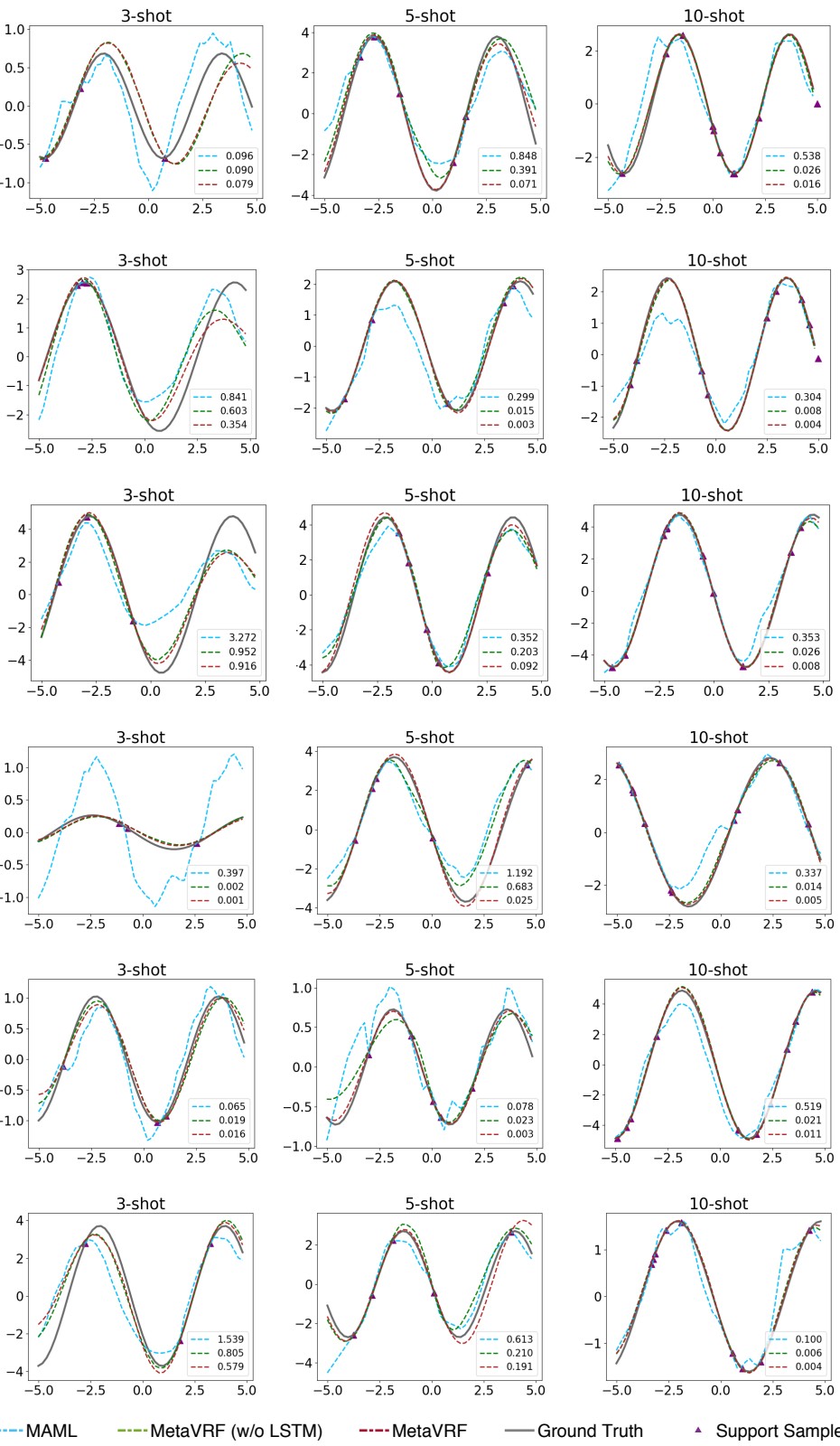

Figure 6:  More results of few-shot regression.

