# OpenReview forum: "Learning to Learn Kernels with Variational Random Features"
_ICLR.cc/2020/Conference — Reject_

### Official Review · AnonReviewer1 · 2019-10-21
**Official Blind Review #1**

**Rating:** 6

**Review:**

This paper proposes a meta-learning framework for learning adaptive kernels using a meta-learner. For representing kernels, the paper learns a variational posterior for the kernel features, by maximizing the Evidence lower Bound. Furthermore, to plug the kernel learning into the meta-learning framework, they let the variational feature posterior to condition on the current support set for adapting and to use a modified LSTM network for accumulating information. Empirically, they compare the proposed MetaVRF with multiple baselines in the standard fewshot classification benchmarks and demonstrate superior performance. They also illustrate that their adaptively-learnt Fourier feature outperforms the standard variational Fourier features.

Strengths,
1, The idea of learning kernels in meta-learning is interesting. In fact, learning a kernel is equivalent to learning a distance between objects. If a reasonable distance between objects can be learnt, using the corresponding kernel should be able to achieve superior performance even if the kernel doesn't adapt in each episode.
2, The proposed method achieves competitive performances. In particular, Figure 5 shows how the performance changes when the test-shot and test-way are varied. It seems surprising that the MetaVRF achieves >90% accuracy for 100-way test when trained on only 5-way 5-shot.

Weakness,
1, The notations in the paper are not well presented. (a) In eq(2), the formula $alpha^t=\Lambda(\Phi^t(x), y)$ is not exact, cuz $\alpha^t$ should depend on the whole support set $S^t$ while $(x,y)$ is only one instance in $S^t$.  (b) In eq(11), the variational posterior $q(w | S^t, w^{1:t-1})$ is not exact either. Because $w^{1:t-1}$ are random variables, they cannot be observed and cannot be conditioned on. Similar issues also exist in the caption of Figure 1.
2, The paper doesn't introduce what is the likelihood $log p(y| x, S, w)$. It is unclear how the kernel regression is adopted in classification.
3, The meta-prior $p(w| x, S)$ depends on the feature of the query point, which doesn't seem to be a common practice in variational inference. It would be beneficial if the authors could explain this and probably validate it empirically.
4, The motivations of the modified LSTM should be clarified more. Does the paper remove h_t in LSTM for removing short-term memory ? Why $\hat{c}_t$ only depends on $e^t$ instead of $[e^t, c^{t-1}]$ ?
5, The paper compares with multiple competitive baselines. However, the settings for the baselines should be better presented. For example, it is strange that the numbers of SNAIL are different with the numbers in their paper. And SNAIL on Omniglot is not reported. Furthermore, another competitive method TADAM (Oreshkin et. al., 2019) should also be compared with.

**Experience Assessment:**

I have published one or two papers in this area.

**Review Assessment: Checking Correctness Of Derivations And Theory:**

I carefully checked the derivations and theory.

**Review Assessment: Checking Correctness Of Experiments:**

I carefully checked the experiments.

**Review Assessment: Thoroughness In Paper Reading:**

N/A

---

> ### Author Response · Authors · 2019-11-10
> **Thank you for your comments.**
>
> 1. Thank you for your careful review. We have updated the second term in eq. (2) with $\alpha^t = \Lambda\left(\Phi^{t}(X), Y\right)$, where $S^t = \{ X, Y \}$, which is what we implemented. We also replaced $\omega^{1:t-1}$ with $\omega^{t}$ in Section 3.2 and Figure 2. The current base $\omega^{t}$ is conditioned on $\omega^{t-1}$ rather than $\omega^{1:t-1}$. In our implementation, the current LSTM cell receives the previous state $c$ and combines it with the input to infer the adaptive spectral distribution, which is consistent with the updated notation now.
>
>
> 2. We now explain the likelihood $\log p(y|x,S,\omega)$, which is a predictive conditional log-likelihood. $y$ is the output random variable, whose distribution is a conditional distribution on the input variable $x$, the data from the support set $S$ and the random Fourier base $\omega$. In our implementation, we obtain $\alpha$ by solving the kernel ridge regression (KRR) in (eq. (3)) in which kernel is computed by using the random bases $\omega$ and the support set $S$. Then we apply eq. (4) for the prediction of $y$ which is obtained by applying the softmax operation. In the optimization, we use the cross-entropy loss.
>
>
> 3. Thank you for this insightful comment. Yes, the meta-prior is slightly different from common practice in variational inference, e.g., variational auto-encoder (VAE). Since our task is supervised learning, the target $y$ to be predicted is conditioned on its input $x$ and the support set, we therefore makes the prior also conditioned on the input data and the support set, rather than using an uninformative prior $N(0,I)$. This is actually similar in spirit to the practice of conditional variational auto-encoder (CVAE).
>
> In our implementation, we choose the permutation-invariant instance-pooling operation (Zaheer et al.,2017) to process the support set into a single vector. Given a data vector $x$, these two vectors are concatenated as the input of the prior network, which outputs the mean and variance for the prior distribution.
>
> 4. Thank for your this insightful comment. The motivations of the modified LSTM are mainly in two folds: (i) to better explore the task dependence and (ii) to implement efficiently.
>
> We're glad to re-clarify the last paragraph in Section 3.2 and all operations in eq .(13).
> During inference, the cell state $c$ stores and accumulates the shared knowledge which is updated for each task throughout the course of learning.
> We simplify LSTM by removing the short-term memory and mainly rely on the "forget-remember" operations to accumulate the shared knowledge from a series of episodes. In eq. (13), the forget gate layer (the 1st line) and input gate layer (the 2nd line) in our LSTM can help refine the cell state (the 4-th line) and gain experience from a batch of tasks. After updating $c$, the task-specific information $e$ combined with the shared knowledge in $ c$ are used for inferring the adaptive spectral distribution (the 5-th line). We use $[ e]$ rather than $[ e, c^{t-1}]$ in the 3-rd line, because we consider the operation as a non-linear mapping of the input, which is unnecessary to use the previous state. In addition, removing the short-term memory can simplify LSTM structure and then promote efficiency during inference. We tried the regular LSTM initially and found our modified version performs better in our tasks.
>
>
> 5. Thank you for the suggestions on more details and results of baselines. For SNAIL, we use the number from Bertinetto et al., 2019) , in which the result of SNAIL is obtained using similar shallow networks as ours. In the original work of SNAIL, they use the deep ResNet-12 embedding (with larger scale filters from 64C to 256C) for miniImageNet and therefore their results are not directly comparable. Actually, we tried to implement SNAIL but we didn't find the official code from the authors, and we adopted the top rate code on Github (https://github.com/eambutu/snail-pytorch). With this code, we trained the SNAIL model with the same embedding network as ours but was not able to achieve the reasonable results (5-way 1-shot: 39.8$\%$; 5-way 5-shot: 55.7$\%$ on miniImageNet). Our finding is consistent with that in Bertinetto et al., 2019) (Here, we cite the sentence from Bertinetto et al. (2019): "Moreover, it is paramount for SNAIL to make use of such deep embedding, as its performance drops significantly with a shallow one." ).
>
> We have added the comparison with TADAM. To make relatively fair comparison, we compare TADAM with our result by using pre-trained embeddings since it uses the ResNet-12 as the backbone embedding. Detailed settings and comparison results are added in Appendix A.

---

> > ### Comment · AnonReviewer1 · 2019-11-13
> > **Thanks for your detailed rebuttal**
> >
> > Thanks for your detailed rebuttal, I think it resolves most of my questions. But I am still kind of skeptical with respect to the prior. Conditioning on $S^t$ is more similar to CVAE, while I don't see the intuitive explanation of conditioning on $x$.

---

> > > ### Author Response · Authors · 2019-11-14
> > > **Thank you for your feedback again.**
> > >
> > > We are very glad to hear that our responses resolve most of your questions.
> > >
> > > We now would like to further explain the meta prior $p(\omega| x, S)$. We show technically in the derivation of the meta ELBO (eqs. 14-17 in the appendix) how the meta prior is conditioned on the input $x$, from which we provide some intuitive explanation to hopefully make it clear to you. The derivation starts with the conditional predictive log-likelihood $\log p(y|x,S)$. That is, we would like to estimate the probabilistic distribution of the target $y$ of the input $x$, and we need to condition on the input data and as well as the context that is the support set $S$. We introduce the latent variable $\omega$ which is the random base in our case. $\omega$ is used to generate the random features for $x$ and therefore should be dependent on $x$ and we further make it conditional on $S$ to leverage the support set under the meta-learning setting. This gives rise to the conditional distribution $p(\omega|x,S)$, based on which the conditional predictive log-likelihood can be re-written as (14). By introducing the variational distribution $q(\omega|S)$ and applying the Jensen's inequality, we achieve the meta ELBO in eq. (17). In analogy to the ELBO in conventional variational inference, we name $p(\omega|x,S)$ as the meta prior. Note that in the variational distribution $q(\omega|S)$ we also make $\omega$ condition on the support set $S$ to leverage the meta-learning setting.  Maximizing the meta ELBO is to minimize the KL between the variational distribution $q(\omega|S)$ and the meta prior $p(\omega|x,S)$, which encourages the model to extract information from the support set for the representation of $x$ in terms of random base $\omega$. By optimizing over samples in the query set, the obtained variational distribution $q(\omega|S)$ is able to infer from the support set the distribution over the base $\omega$ that can generate informative random features for each sample $x$ in the query set. In addition, the meta prior is the conditional distribution on the input data $x$ can also be shown from the perspective of the minimization of the KL divergence between variational distribution $q(\omega|S)$ and the posterior $p(\omega|x,y,S)$ (eqs. (18)-(23)).

---

> ### Author Response · Authors · 2019-11-10
> **Thank you for your comments.**
>
> Thanks for your insightful review and constructive comments.  We especially thank you for your careful and detailed reviews on the notations, which helps make our presentation more precise.
>
> Thank you for acknowledging that the idea of learning kernels in meta-learning is interesting. Yes, we totally agree with you on that learning a kernel is equivalent to learning a distance between objects. Indeed, our inference of spectral distributions depending on also previous episodes (leveraging shared knowledge across related tasks) is to learn such a powerful kernel that can provide a reasonable distance between objects. The promising performance on the versatility experiments with inconsistent settings between training and test also, to some extent, demonstrates the effectiveness of learning a power kernels by exploring related tasks for few shot learning.
>
> Our detailed responses to your questions are provided below.

---

### Official Review · AnonReviewer2 · 2019-10-26
**Official Blind Review #2**

**Rating:** 6

**Review:**

This paper studies meta-learning problem with few-shot learning settings. The author proposes a learn each task predictive function via the form of random Fourier features, where the kernel is jointly learned from all tasks. The novel part is the parametrization of inference network using LSTM such that the random feature samples of t-th task conditional depending on all previous task 1,...,t-1, which is an interesting way of modeling kernel spectral distribution. The experiment results show improvement of the proposed methods compared to SoTA meta learning algorithms.

In general, the writing of the paper is clear, and the proposed method is interesting and novel. However, there are parts missing in the experiment setting.  I would love to increase my score if the author could address the following questions/comments:
(1) How do you choose the meta prior distribution? It should be a basic kernel family such as RBF Gaussian or mixture of RBF?
(2) In Table 1 and Table 2, the benefit of using LSTM only gives very marginal improvement over w/o LSTM. Are the results statistically significant?
(3) The experiment missed the simple kernel learning baseline, such as kernel alignment [1] and its variants [2]. If using these task-independent way to do kernel learning, what’s their performance compared to you proposed method?
(4) When learning the RFF spectral distribution using LSTM over a sequence of tasks, does the order of task matter?


[1] Learning kernels with random features, NIPS 2016.
[2] Implicit kernel learning, AISTATS 2019.


**Experience Assessment:**

I have published one or two papers in this area.

**Review Assessment: Checking Correctness Of Derivations And Theory:**

I assessed the sensibility of the derivations and theory.

**Review Assessment: Checking Correctness Of Experiments:**

I assessed the sensibility of the experiments.

**Review Assessment: Thoroughness In Paper Reading:**

I read the paper at least twice and used my best judgement in assessing the paper.

---

> ### Author Response · Authors · 2019-11-10
> **Thank you for your comments.**
>
> Thank you for your insightful review and constructive comments. Thank you for pinpointing the novel part of this work: using an LSTM based inference network for kernel learning with random features depending on previous tasks.
>
> 1. Yes, we choose the meta prior distribution to be the spectral distribution of the basic Gaussian RBF family. Since the spectral distribution is Gaussian, we can conduct the inference efficiently by inferring $\mu$ and $\sigma$, while achieving highly representational kernels due to the strong nonlinear learning capability of the Gaussian kernel. Thank you for this comment. Yes, it is a great idea to use a mixture of Gaussian, which would be able to generate more informative kernels compared to a single Gaussian. We would like to explore it in our future work.
>
> 2. Thank you for this comment. The improvements by using LSTM are statistically significant, which is based on a large number of runs (3,000) on each dataset. On the Omniglot dataset, the performance of most methods saturates, with the accuracy over $99\%$, which would explain that the improvement using LSTM is slight on this dataset. On other datasets, such as miniImageNet, on the 5-way 1-shot task, the improvement is relatively large from 52.9$\%$ to 54.3$\%$.
>
> 3. Thanks for this insightful comment. Indeed, kernel alignment offers a nice principle for kernel learning, which has been proven useful in conventional learning tasks [1][2]. However, it would not be directly applicable to few shot learning tasks by learning kernels using task-independent way due to that only a few training samples is available in each task (only one sample in one-shot learning) for training. This makes it hard, if not impossible, to learn a reasonable kernel for the task.
>
> Actually, the our model can incorporate the kernel alignment principle into the objective for the base-learner, that is, we add a kernel alignment term in conjunction with the cross-entropy loss. We tried this in our original experiment, while we did not observe any performance gain by using kernel alignment in our experiments. We conjecture that it is because the cross-entropy loss is already powerful enough in this scenario.
>
> We would also like to add that our method is fundamentally different from [1][2] though we are all kernel learning based on random features.
>
> (1) We learn to infer the spectral distributions from data of a specific task while exploring dependency of a set of related tasks. [1][2] learn an optimal configuration, i.e., weights of random bases, of random features, where the bases are drawn from the fixed spectral distribution.
>
> (2) Our method is a one-stage learning, for few shot recognition tasks, while [1][2] are in a two-stage learning way, for conventional learning tasks, where the kernel is learned in a separate prior stage.
>
> 4. Thank you for this comment. The order of tasks does not matter in our model. We mainly leverage the "remember-forget" mechanism in LSTM to accumulate and refine the shared knowledge from a sequence of tasks. In our experimental implementation, we randomly sample a bunch of episodes from training data in each iteration, which also makes the order of tasks not matter.

---

### Official Review · AnonReviewer3 · 2019-10-28
**Official Blind Review #3**

**Rating:** 8

**Review:**

The paper focuses on the topic of meta-learning for few-shot learning and explores kernel approximation with random fourier features for this problem. The authors propose to learn adaptive kernels by meta variational random features, and evaluate their approach on different few-shot learning tasks, comparing it against recent meta-learning algorithms.

The paper is well-motivated and well-written. On page 8, the authors mention related works that were not included for comparison because they rely on pre-trained embeddings or large-scale deep architectures. It would have been interesting to see the difference in performance.

**Experience Assessment:**

I do not know much about this area.

**Review Assessment: Checking Correctness Of Derivations And Theory:**

I assessed the sensibility of the derivations and theory.

**Review Assessment: Checking Correctness Of Experiments:**

I assessed the sensibility of the experiments.

**Review Assessment: Thoroughness In Paper Reading:**

I read the paper at least twice and used my best judgement in assessing the paper.

---

> ### Author Response · Authors · 2019-11-10
> **Thank you for your comments.**
>
> Thank you for your insightful review and very supportive comments.
>
> We're very glad to receive your comment that our work is well-motivated. Indeed, it was a gap to explore kernels that is proven a powerful tool in conventional learning scenarios for few shot learning under the meta-learning framework. Motivated to fill this gap, we propose learning adaptive kernels in the meta-learning setting for few shot learning, where we formulate it as a conditional variational inference problem. Moreover, our kernel learning based on random Fourier features is achieved by inferring task-specific spectral distributions while exploring task dependency by using an LSTM based inference network.
>
> Thank you for your great suggestion. For your interest, we implement with pre-trained embeddings for comparison to see the difference in performance. Specifically, we use the pre-trained embedding extracted from widen residual networks (WRN-28-10) as used in (Rusu et al.,2019). As expected, the performance is improved over that trained from scratch. This is reasonable since features based on large-scale pre-trained embedding can be more informative. We have also made a head-to-head comparison with LEO (Rusu et al.,2019) using the same pre-trained embedding on the miniImageNet dataset. Our MetaVRF performs better than LEO, especially on the 5-way-1-shot task, which again shows the effectiveness of our MetaVRF. Detailed experimental settings and results have been added in Appendix A.

---

### Author Response · Authors · 2019-11-10
**Brief summary**

First of all, we would like to thank all reviewers for their insightful reviews, supportive comments and great suggestions.

We summarize our major updates as follows:
1. We have added more experimental results and comparison with other methods, e.g., SNAIL and TADAM, which include the result using pre-trained embeddings and deep architectures. More comparison results are added in Table 3 associated with discussion in Appendix A.
2. We have updated the notations in eq. (2) and eq. (11) in Section 3.2 and in the caption of Figure 2.
3. We have added detailed settings for SNAIL in the penultimate paragraph of Section 5.2. The result of SNAIL on Omniglot has been added in Table 2.

---

### Decision · Program_Chairs · 2019-12-19

**Decision:**

Reject

**Comment:**

The paper looks at meta learning using random Fourier features for kernel approximations. The idea is to learn adaptive kernels by inferring Fourier bases from related tasks that can be used for the new task. A key insight of the paper is to use an LSTM to share knowledge across tasks.

The paper tackles an interesting problem, and the idea to use a meta learning setting for transfer learning within a kernel setting is quite interesting. It may be worthwhile relating this work to this paper by Titsias et al. (https://arxiv.org/abs/1901.11356), which looks at a slightly different setting (continual learning with Gaussian processes, where information is shared through inducing variables).

Having read the paper, I have some comments/questions:
1. log-likelihood should be called log-marginal likelihood (wherever the ELBO shows up)
2. The derivation of the ELBO confuses me (section 3.1). First, I don't know whether this ELBO is at training time or at test time. If it was at training time, then I agree with Reviewer #1 in the sense that $p(\omega)$ should not depend on either $x$ or $\mathcal {S}$. If it is at test time, the log-likelihood term should not depend on $\mathcal{S}$ (which is the training set), because $\mathcal S$ is taken care of by $p(\omega|\mathcal S)$. However, critically, $p(\omega|\mathcal S)$ should not depend on $x$. I agree with Reviewer #1 that this part is confusing, and the authors' response has not helped me to diffuse this confusion (e.g., priors should not be conditioned on any data).
3. The tasks are indirectly represented by a set of basis functions, which are represented by $\omega^t$ for task $t$. In the paper, these tasks are then inferred using variational inference and an LSTM. It may be worthwhile relating this to the latent-variable approach by Saemundsson et al. (http://auai.org/uai2018/proceedings/papers/235.pdf) for meta learning.
4. The expression "meta ELBO" is inappropriate. This is a simple ELBO, nothing meta about it. If we think of the tasks as latent variables (which the paper also states), this ELBO in equation (9) is a vanilla ELBO that is used in variational inference.
5. For the LSTM, does it make a difference how the tasks are ordered?
6. Experiments: Figure 3 clearly needs error bars, and MSEs need to be reported with error bars as well;
6a) Figures 4 and 5 need error bars.
6b) Error bars should also be based on different random initializations of the learning procedure to evaluate the robustness of the methods (use at least 20 random seeds). I don't think any of the results is based on more than one random seed (at least I could not find any statement regarding this).
7. Table 1 and 2: The highlighting in bold is unclear. If it is supposed to highlight the best methods, then the highlighting is dishonest in the sense that methods, which perform similarly, are not highlighted. For example, in Table 1, VERSA or MetaVRF (w/o LSTM) could be highlighted for all tasks because the error bars are so huge (similar in Table 2).
8. One of the things I'm missing completely is a discussion about computational demand: How efficiently can we train the model, and how long does it take to make predictions? It would be great to have some discussion about this in the paper and relate this to other approaches.
9. The paper evaluates also the effect of having an LSTM that correlates tasks in the posterior. The analysis shows that there are some marginal gains, but none of the is statistically significant. I would have liked to see much more analysis of the effect/benefit of the LSTM.

Summary: The paper addresses an interesting problem. However, I have reservations regarding some theoretical bits and regarding the quality of the evaluation. Given that this paper also exceeds the 8 pages (default) limit, we are supposed to ask for higher acceptance standards than for an 8-pages paper. Hence, putting everything together, I recommend to reject this paper.